# Ground Beetles (*Carabidae*) in the Short-Rotation Coppice Willow and Poplar Plants—Synergistic Benefits System

**Natalia Stefania Piotrowska [1,*], Stanisław Zbigniew Czachorowski [1] and Mariusz Jerzy Stolarski [2]**

[1] Department of Ecology and Environmental Protection, Faculty of Biology and Biotechnology, University of Warmia and Mazury in Olsztyn, Plac Łódzki 3, 10-727 Olsztyn, Poland; stanislaw.czachorowski@uwm.edu.pl

[2] Department of Plant Breeding and Seed Production, Faculty of Environmental Management and Agriculture, University of Warmia and Mazury in Olsztyn, Plac Łódzki 3, 10-724 Olsztyn, Poland; mariusz.stolarski@uwm.edu.pl

\* Correspondence: natalia.piotrowska@uwm.edu.pl

**Abstract:** In a short period, we have observed the rapid expansion of bioenergy, resulting in growth in the area of energy crops. In Europe, willow and poplar growing in short-rotation coppices (SRC) are popular bioenergy crops. Their potential impact on biodiversity has not yet been fully investigated. Therefore, there are many uncertainties regarding whether commercial production can cause environmental degradation and biodiversity impoverishment. One of the aspects examined is the impact of these crops on entomofauna and ecosystem services. The best-studied insect group is ground beetles from the *Carabidae* family. This work gathers data on biodiversity and the functions of carabids in willow and poplar energy plants. The results of these investigations show that energy SRC plants and *Carabidae* communities can create a synergistic system of mutual benefits. Willow and poplar plants can be a valuable habitat due to the increased biodiversity of entomofauna. Additionally, SRC creates a transitional environment that allows insect migration between isolated populations. On the other hand, ground beetles are suppliers of ecosystem services and make a significant contribution to the building of sustainable agriculture by pest control, thereby ameliorating damage to field crops.

**Keywords:** willow SRC; energy plants; ground beetles; *Carabidae*; ecosystem services; invertebrate biodiversity

## 1. Introduction

Currently, in the EU28 (28 European Union countries), the acreage of lignocellulosic plants is estimated to be 50,000 hectares of short-rotation coppices (SRC), mainly willow (*Salix* spp.) and poplar (*Populus* spp.) plants [1]. However, their impact on biodiversity remains only partially known [2]. It is considered that treatment for the natural environment differs for different types of energy crops [3–5].

Non-edible lignocellulosic plants, including willow and poplar, are used for heat and power generation and second-generation liquid biofuel production. It is considered that use of lignocellulosic plants can reduce competition with traditional crops for land and water resources [6], especially when grown on marginal land unsuitable for food production [7]. Despite this, many uncertainties exist about the potential impacts of biomass crops on the environment and biodiversity. There is major concern that commercial production can cause environmental degradation,

significantly raising the risk of habitat fragmentation, native extinction, and bio-invasion [6]. Some studies report that the large-scale homogeneous landscape of biofuel plantation has resulted in a simplified bio-community and food web, severely damaging ecosystem services and contributing to the decline in the biodiversity, in particular in areas of high nature-conservation value [8]. Therefore, although biomass can help as an energy source to reduce the world's reliance on fossil energy and mitigate global warming [9], there is growing concern about the hypothetical disturbances biofuel can have on ecosystems and biodiversity. Accordingly, in this article, we will focus on willow and poplar SRC. The impact of these plantations on abiotic factors is well understood [10–36], but they can also affect the biotope, changing the species composition of plant and animal communities that directly inhabit the cultivation area, as well as adjacent habitats. So far, however, no comprehensive research has been made that fully shows the impact of willow plantations on the natural environment. It is known that in large agrocenoses, the introduction of small environmental patches of energy crops into cultivation can contribute to an increase in landscape mosaicism, significantly reduced by monocultures [7,37,38].

Studies carried out on large-scale monoculture crops show that the decrease in landscape mosaicism causes many adverse changes at all levels of the trophic chain. These changes occur both at the local level, limited to populations living in a given area, and global, affecting the structure of whole biomes. Over the past few decades, a significant decline in total biomass and the diversity of insect clusters has been observed, especially in North America and Europe [39–41]. Recent research has indicated that flying insect biomass decline may be up to 75% in some areas [42]. The main reasons hypothesized are anthropogenic drivers including land-use change [43,44], transport routes [45–47], environmental pollution, and pesticides [44,48–50] as well as climate changes [43,44,51,52]. Since the end of World War II, we have observed great intensification of agriculture and the evolution of the entire agrocenoses. Crop intensification and excessive pesticide use and accompanying processes, such as melioration or cessation of grazing, have led to the degradation of the habitat [53]. Those changes have contributed to the dramatic loss of biodiversity—many organisms have lost their ecological niches because of lack of shelter or other environmental resources, such as nutrition base. Thus, progressive degradation and the breakdown of ecological networks has been observed.

Due to the possibility of negative environmental effects, research has also been carried out on the influence of energy crops on the fauna that inhabits them. These studies have mainly focused on birds [34,54–67], and a smaller number of them concern mammals [60,61,68–72] or invertebrates [73–93]. Experiments have also been carried out to investigate the differences between biodiversity in woody crops and herbaceous perennial crops and grasses, such as, e.g., Virginia mallow or miscanthus [94]. The most investigated insect group in energy crops are carabids, the largest family of adephagan beetles (Coleoptera: *Carabidae*) [95,96]. So far, more than 40,000 species of ground beetles have been described, including more than 2700 in Europe and over 2000 in North America [97]. Due to their high plasticity, ground beetles have acquired a wide variety of habitats. This group includes eurytopic (ubiquitous), forest (occurring in wooded environments), open area (found in fields and meadows), coastal (associated with wetlands and banks of waters), and peat bog species. They differ in preference regarding humidity of environment, development cycle, size, and eating habits. Due to the above, *Carabidae* can be divided into five main groups: large predatory species (body length over 12 mm), medium predatory species (5–12 mm), small predatory species (<5 mm), hemizoophages (half-herbivorous), and phytophages (herbivorous) [85,86,92,98,99]. Most beetles in this group are characterized by a high level of predation. Although a diversified forest ecosystem is abundant in factors reducing the presence of phytophages, highly specialized agrocenoses are exposed to an excessive increase in the number of pests. As a result, in addition to anthropogenic factors, beetles from the *Carabidae* family are one of the main groups that contribute to the control of the pest population. Moreover, their services are not limited to SRC but are also provided to adjacent crops, which plays an important role in sustainable agriculture [100]. Accordingly, the role they are playing in ecological services cannot be underestimated.

The dominance structure in the *Carabidae* population may be a reflection of the habitat conditions [92]. Sharpening the structure of dominance can be considered a result of destructive factors existing

in the environment [14,101]. Meanwhile, in stable habitats with the correct structure, smooth transitions are observed between the gradually decreasing percentages of species from individual groups [86]. Similarly, the trophic structure of the carabid population changes depending on the state of their living environment. Previously, the presence of large zoophages was considered the most desirable [102], but now the important role of herbivorous species has also been emphasized [103,104].

According to research by [105] in highly intensified agrocenoses, large zoophages are replaced by smaller predatory beetles, and as the pressure increases, the proportion of granivorous carabids increases in the grouping. However, these studies were carried out for meadows and arable fields, and similar works for woodland habitats such as poplar and willow crops are lacking. Therefore, more investigations should be done to estimate this factor for SRC [92,105]. *Carabidae* were chosen as the subject of this review for several reasons:

-   As a well-known group of epigeic insects, they can be treated as a monitoring group;
-   They play an important role, providing valuable ecosystem services for willow plantations and adjacent crops;
-   Compared to other insect groups for which the amount of data is negligible, there are more studies for *Carabidae* clusters, allowing for analysis.

The novelty of this work is that it show a new view on SRC plantations as environmental islands—areas that can be refugia and environmental corridors for endangered populations. The purpose of this work is not only to show the diversity of entomofauna but also to draw attention to the relationship between the habitat of *Carabidae* and the shape of their population and their ecologic function, investigations of which have not been extensively developed so far. This will allow a better understanding of the role not only of the ground beetles themselves, as a group providing ecosystem services, but also of the entire complex environment of the energy willow plantation.

## 2. Materials and Methods/Data Collection and Selection

This review presents the most important studies on carabids in energy willow and poplar plantations, mainly in Central Europe. The areas covered are presented in Figure 1.

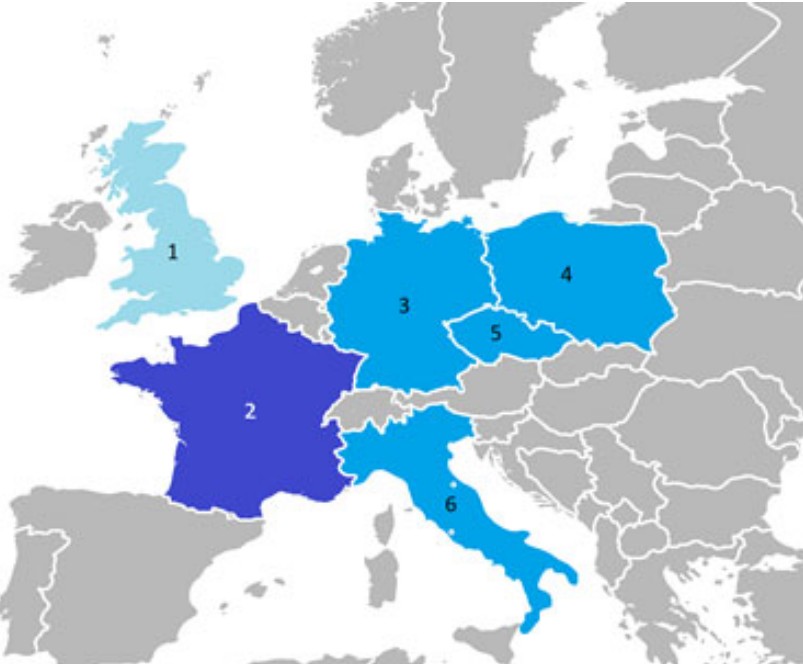

**Figure 1.** Countries for which original research works on *Carabidae* beetle fauna on poplar and willow plants were available in searched databases (1—Great Britain, 2—France, 3—Germany, 4—Poland,

5—Czech Republic, 6—Italy; light blue—works used in the study; blue—works used as well not used in the study; dark blue—works not used in the study).

Materials were acquired from articles published in English, German, and Polish. As the study materials, we used proceedings papers, original research studies, and review articles published between 1950 and 2020, mostly concerning plantations located in Europe. The research was conducted in three scientific databases: International Web of Science, Scopus , and CEON Biblioteka Nauki . In the final database, which is an information source of science papers published only in Polish periodicals, the searched keywords were: "wierzba *Carabidae*", "wierzba owady epigeiczne", "wierzba biegaczowate","wierzba bioróżnorodność owadów", "wierzba bioróżnorodność entomofauny", "wierzba entomofauny", "wierzba owadów", "wierzba biegaczowatych", "wierzby biegaczowatych", "wierzby owadów epigeicznych", "wierzby *Carabidae*", "wierzby bioróżnorodność owadów", "wierzby bioróznorodność entomofauny", "wierzby entomofauna", "wierzby biegaczowate", and "wierzby owady epigeiczne". Meanwhile, from the Web of Science database, the material was acquired by searching for the following words: "energetic willow *Carabidae*", "energy crops *Carabidae*", "*Salix viminalis Carabidae*", "short rotation coppice *Carabidae*", "energetic poplar *Carabidae*", "energetic willow ground beetles", "energy crops ground beetles", "*Salix viminalis* ground beetles", "short rotation coppice ground beetles", "energetic poplar ground beetles", "energetic willow epigeic insects", "energy crops epigeic insects", "*Salix viminalis* epigeic insects", "short rotation coppice epigeic insects", "energetic poplar epigeic insects", "energetic willow insects biodiversity", "energy crops insects biodiversity", "*Salix viminalis* insects biodiversity", "short rotation coppice insects biodiversity", and "energetic poplar insects biodiversity". In Scopus, to restrict the search, we used words such as: "energetic willow *Carabidae*", "*Salix viminalis Carabidae*", "short rotation coppice *Carabidae*", "energetic poplar *Carabidae*", "energetic willow ground beetles", "*Salix viminalis* ground beetles", "short rotation coppice ground beetles", "energetic poplar ground beetles", "energetic willow epigeic insects", "*Salix viminalis* epigeic insects", "short rotation coppice epigeic insects", "energetic poplar epigeic insects", "energetic willow insects biodiversity", "*Salix viminalis* insects biodiversity", "short rotation coppice insects biodiversity", and "energetic poplar insects biodiversity".

In both databases, records were searched in all fields. In the Web of Science, our investigations included all collections. To broaden the results of the research, the "References" sections of the research articles used to prepare this review were studied, and relevant articles, if were not previously included, were added to our investigations. Furthermore, similar proceedings were applied to articles found in the Scopus database, where the "cited by" section was used. Most of the articles found were excluded from the analysis due to low relevance and duplications. The exclusion criteria for the selected articles were: no connection with the subject, the articles concerned other species of energy plants, the articles related to animals other than *Carabidae*, and the articles described plantations from outside of Europe.

The CEON Biblioteka Nauki database showed 58 records, among which only 3 records were selected as related to the topic. The International Web of Science database indicated 77 records, among which 22 were chosen as relevant. The Scopus database yielded the highest number of articles, as many as 5517 records. Due to this, it was decided to narrow down the search area and to reject key phrases containing "energy crops". This reduced the number of records to 458, out of which 24 relevant results were selected.

The articles collected were sorted according to the year of publication and the country of origin (the country was assigned based on the corresponding author). The results are presented in Figures 2 and 3. In the years 1998–2020, 32 articles on *Carabidae* biodiversity in energy willow and poplar plantations were published. Most of them, as many as 15 items, were published by German authors. Their number is three times higher than that of Polish and British publications. For Czech and Swedish authors, a database search showed two publications each. One publication was found for the Netherlands, Slovakia, and Belgium, respectively. In investigations for years from the 1950s until 1998, the databases did not show any publications described by the keywords used. In 1998, one

publication was issued, and another was issued after 9 years, in 2008. An upward trend was observed for the following years, with 6 publications issued in 2012. For 2013 and 2014, there were 4 publications each. After this year, there is a clear decrease in the number of publications—in 2015, no publications were issued, in 2016, two were issued, and in the following years, only one publication per year was issued.

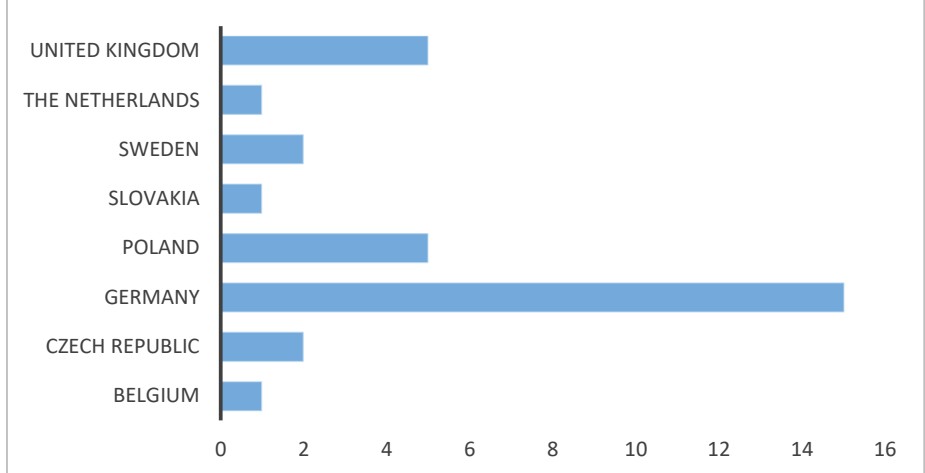

**Figure 2.** Publications on *Carabidae* in energy willow and poplar plantations—number per country. Articles found in three scientific databases: International Web of Science, Scopus, and CEON Biblioteka Nauki after discarding unrelated works and duplicated records.

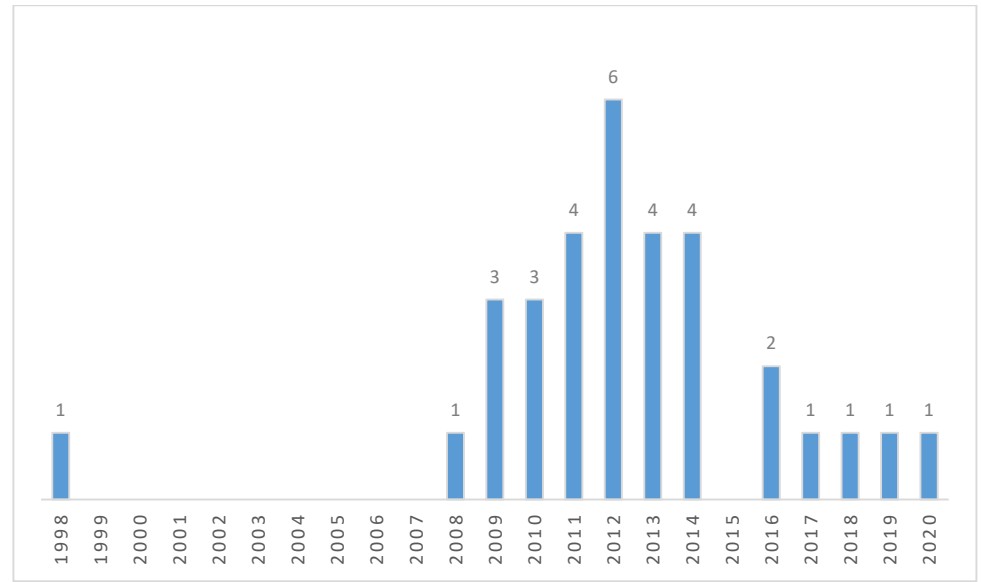

**Figure 3.** Publications on *Carabidae* in energy willow and poplar plantations—number per year. Articles found in three scientific databases: International Web of Science, Scopus and CEON Biblioteka Nauki after discarding unrelated works and duplicated records.

From the publications, four articles were selected that contained sufficient data to create an ecological characterization of *Carabidae* inhabiting the plantations described in them [85–87,92]. This enabled the designation of 12 research areas and the determination of the dominance structure, as well as phenology, hygro- and habitat preference, trophy group and dispersion powers. Energy plants characteristics are described in Table 1. To describe the phenology, hygropreference, trophy group and dispersion powers, individual species were assigned appropriate characteristics, and then their percentage of the total number of species was calculated. To describe the dominance structure

based on abundance, species were assigned to different domination classes: eudominants, dominants, subdominants, recedents and subrecedents. Both methods were used to describe habitat preference, assessing both the percentage share of individual species in the total pool and the percentage of individuals with given environmental preferences on each of the plantations. Additionally, the number of forest species occurring on each study plot was shown concerning the age of the plantation and the size of the stand. The obtained results are presented in the form of tables and graphs in the Results and Discussion sections. Microsoft Office Excel was used for its design.

**Table 1.** Research articles that were the basis of this review–plantations characteristic (Local.—localization, Plant.—plantation, Years—years of investigation).

| References | Investigated Issues | Local. | Plant. Type | Plant. Age | Canopy Age | Adjacent Habitat | Years Date |
|---|---|---|---|---|---|---|---|
| [86] | 1. Species richness and diversity, dominance structure. 2. Community structure: trophic, habitat preference, humidity preference, development type. | Northeast Poland | willow | 8–9 years | Not given | none | 2004–2005 |
| [85] | 1. Species richness and diversity, dominance structure. 2. Community structure: trophic, habitat preference, humidity preference, development type. | Northeast Poland | willow | 1 to 3 years | 1 to 3 years | none | 2005–2006 |
| [87] | 1. Species composition and abundance of ground beetles inhabiting unexploited willows plantation. 2. Population ecological characteristic and dominance structure. 3. Margalef's index, Shannon' diversity, Evenness H/log(N) Pielou. | Southeast Poland | willow | not given | 8 and 9 years | none | 2011–2012 |
| [92] | 1. Species richness and diversity, dominance structure. 2. The similarity to natural woodlands. | Po Valley, Italy | poplar | 1 to 10 years | not given | natural woods, crops: maize, tobacco | 1989–1999 |
| [102] | 1. Influence of the plantation vicinity and anthropogenic factors on *Carabidae* assemblages. 2. The structure of beetle communities. | South Bohemia (Czech Republic) | willow | 2, 4, 6, 8 years | not given | Field with a stream; pond, field, alder trees and meadow; pasture, field, cultural forest; | 2007 |

| | | | | | | |
|---|---|---|---|---|---|---|
| [81] | 1. Comparison of how predation processes by ground arthropods varied between short rotation coppice (SRC) willow bioenergy plantations and alternative land-uses: arable and set-aside. 2. Predation pressure investigations: prey removal assay coupled with pitfall traps and direct searches | North Nottinghamshire, England | willow | 1 to 10 years | 1 to 9 years | set-aside, arable | 2008 |
| [88] | 1. Simpson biodiversity index, evenness, level of anthropogenic influence 2. The influence of the length of rotation on biodiversity parameters; | Peklov, Czech Republic | poplar | 9 years | 1, 3, 6 years rotations | none | 2003–2008 |
| [89] | 1. Taxonomy and identification. 2. Species traits and categorization. 3. Habitat preferences. 4. Endangered species. 5. Dispersal of forest species; corridor function. 6. Species traits concerning age of the SRC versus age of the SRC standing crop. 7. Factors influencing SRC biodiversity functions. | Germany (different sites) and the Czech Republic | willow poplar | 1 to 23 years | 1 to 9 years | different | Meta-study |

## 3. Results

Based on the data analysis, presented in Figure 4, the predominance of *Carabidae* species preferring the environment of open areas was found. The comparison of Figure 5 and Table 2 showed that the percentage share of species preferring open areas is independent of the age of the plantation, while its dependence on the age of the stand seems impossible to assess due to insufficient data. However, their share appears to decline as the age of the canopy increases. Species preferring open areas also constituted the most numerous group in terms of the number of individuals in a given population, which is presented in Figure 5. This tendency occurred on most plantations, both willow [85–87] and poplar [92]. Species with unknown habitat preference constituted less than 15% of all recorded species. On three plantations, the niche of open ground species was occupied by eurytypical species. The share of forest species was greater, at over 18 percent (Figure 4). The number of forest species depending on the age of the plantation and the age of the canopy is presented in Table 2. In some plantations, there was a visible predominance of ground beetles representing forest species, which, however, was not correlated with the age of the plantation. Unfortunately, the amount of data allowing us to assess the influence of the surrounding environment on the number of *Carabidae* from different ecological groups was insufficient. A similar correlation concerning the age of the stand is impossible to analyze due to the lack of sufficient data, as the analyzed publications lack information enabling its determination. The number of peatland species was less than 10% that of all species collected, and the number of individuals with this preference was even lower.

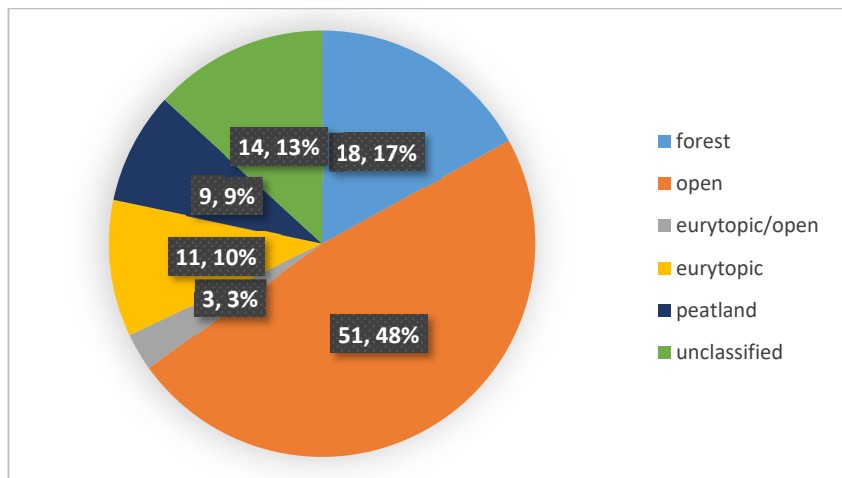

**Figure 4.** Percentage distribution of *Carabidae* species found in selected willow and poplar plantations depending on environmental preferences. The diagram was made based on selected publications, including data on the ecological characteristics of *Carabidae* communities inhabiting investigated plants.

**Table 2.** The number of *Carabidae* species preferring the forest environment, taking into account the number of species in the classes of eudominants (ED) and dominants (D). The table was made based on selected publications, including data about the ecological characteristic of *Carabidae* communities inhabiting investigated plants (W1, W2 and unmarked–willow plots; P1–P4–poplar plots).

| Investigated Plots and Year of Investigation | Plantation Age (Years) | Canopy Age (Years) | Number of Forest Species (including ED and D) |
|---|---|---|---|
| [86] 2004 | 8 | Not given | 5 (1 ED) |
| [86] 2005 | 9 | Not given | 4 (2 ED) |
| [85] 2005 W1 | 2 | 2. | 2 (2 ED) |
| [85] 2006 W1 | 3 | 3. | 2 |
| [85] 2005 W2 | 1 | 1 | 2 (2 ED) |
| [85] 2006 W2 | 2 | 2 | 2 (1 ED) |
| [87] 2011 | Not given | 8 | 7 (1 ED) |
| [87] 2012 | Not given | 9. | 5 (1 ED) |
| [92] 1989 P1 | 2 | Not given | 5 |
| [92] 1999 P2 | 6 | Not given | 3 (1 ED, 1 D) |
| [92] 1999 P3 | 6 | Not given | 2 (1 D) |
| [92] 1991 P4 | 10 | Not given | 4 (2 ED) |

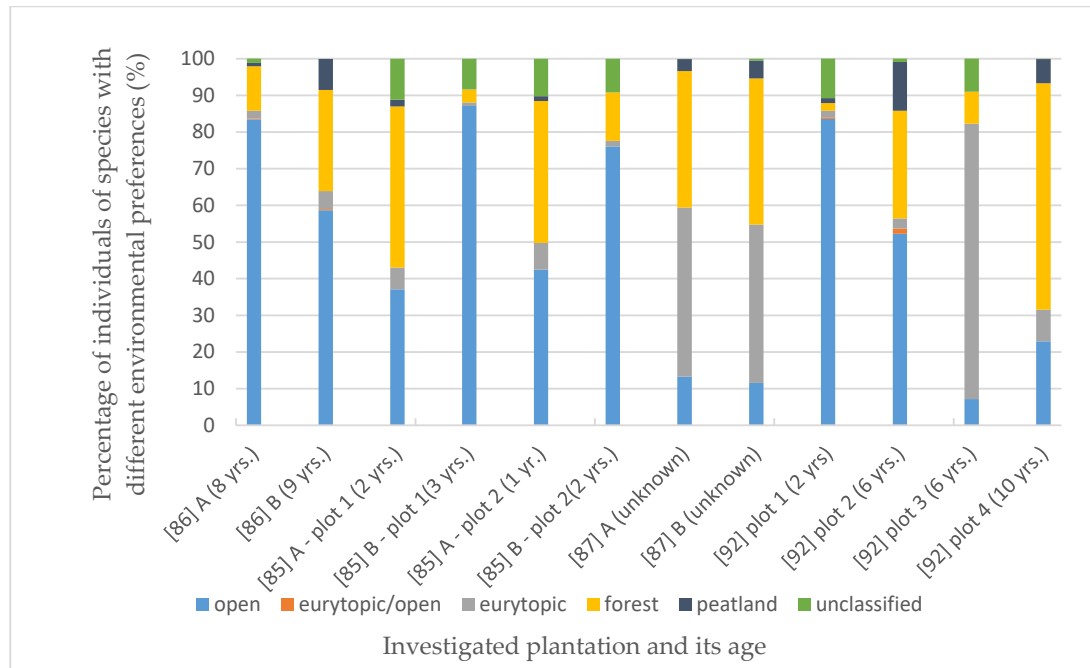

**Figure 5.** Habitat preferences vs. plantation age. Percentage of individuals of species with different environmental preferences depending on the age of the plantation (in brackets). The last four plots, studied by [92], are poplar crops. The remaining 8 are willow plantations [85–87], A—first year of investigations, B—second year of investigations.

A wider assessment of the carabid population in the SCR plantation, taking into account the structure of dominance, trophic structure, and species properties, indicated the predominance of poorly specialized species with wide environmental tolerance. Data analysis showed that the number of brachypterous species was smaller than that of macropterous ones (Figure 6), comprising only one fifth of all examined species. Macropterous species, characteristic for disturbed environments, constituted 64% of the investigated population. Similarly, species with spring biology were significantly more numerous than autumn biology ones, which are characteristic of well-balanced environments (Figure 7). The compilation of the collected data showed that the dominance structure of the studied populations was poorly balanced. In 11 out of 12 assessed plots, the number of eudominants predominated, amounting to over 60% in eight cases, and over 70% of all specimens in five cases. (Figure 8). Granivorous species predominated, followed by small and large zoophages (Figure 9). In terms of hygropreference (Figure 10), mesophiles were the most numerous. A large group was also species with undefined hygropreference. Poor *Carabidae* specialization may indicate an imbalance in the environment in which they live. On the other hand, there were no sharp transitions between individual trophic groups, while the percentage of large zoophages was average. This proves that even though the plantation environment is subject to intensive changes, it is possible to maintain a sustainable biocenosis there.

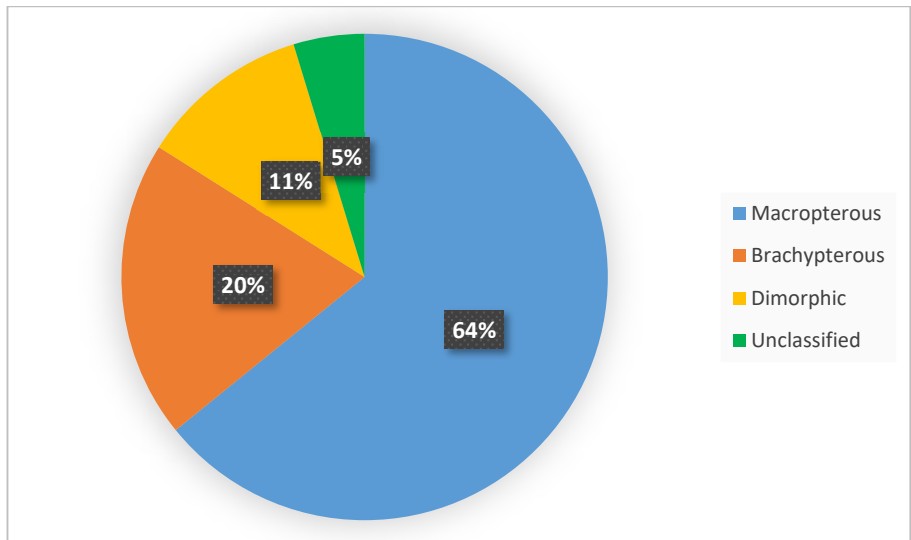

**Figure 6.** Percentage distribution of *Carabidae* species with different dispersibility. This diagram was made based on selected publications, including data on the ecological characteristics of *Carabidae* communities inhabiting the investigated plants.

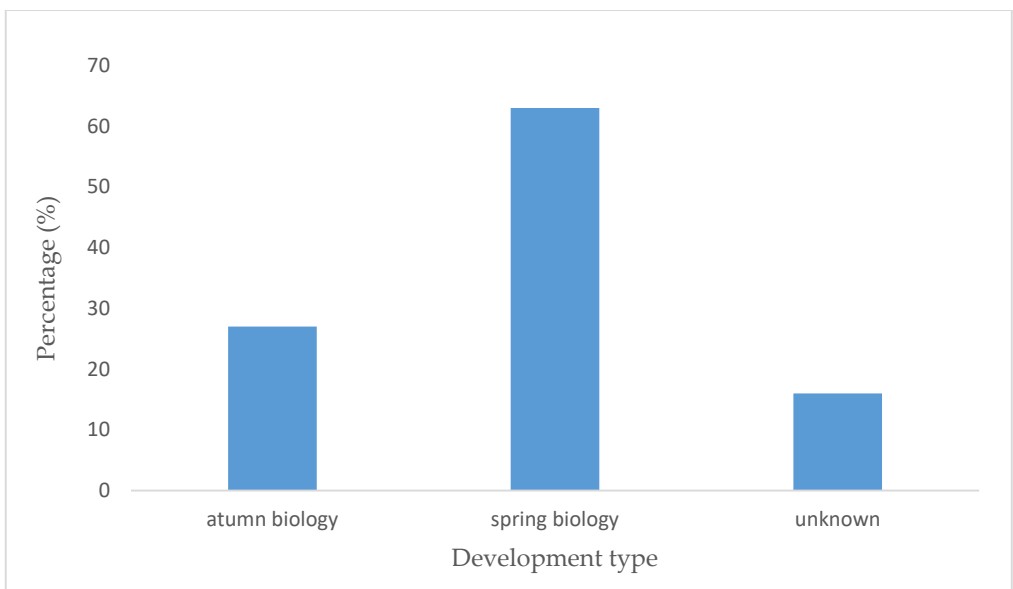

**Figure 7.** Per cent of *Carabidae* species with a different development type. This phenology diagram was made based on selected publications, including data on the ecological characteristic of *Carabidae* communities inhabiting the investigated plants.

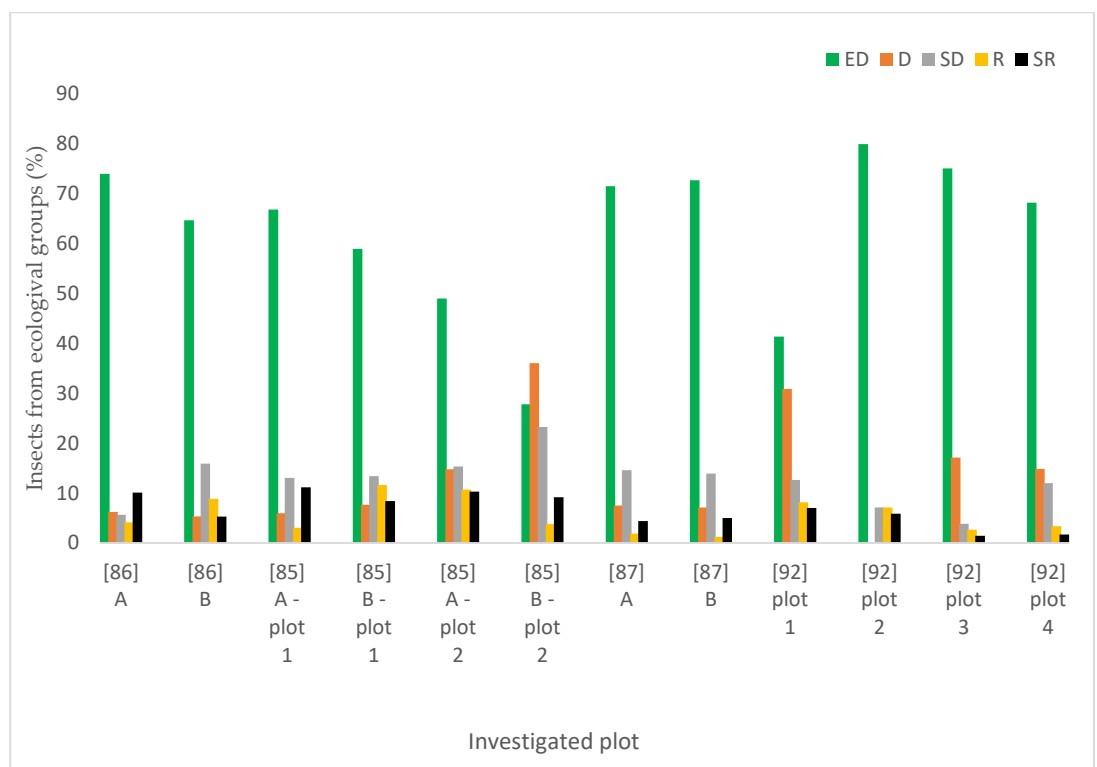

**Figure 8.** Structure of dominance in the *Carabidae* population based on the percentage of insects from particular groups (ED—eudominants; D—dominants; SD—subdominants; R—recedents; SR—subrecedents). This diagram was made based on selected publications, including data on the ecological characteristics of *Carabidae* communities inhabiting the investigated plants. (A—first year of investigations, B—second year of investigations).

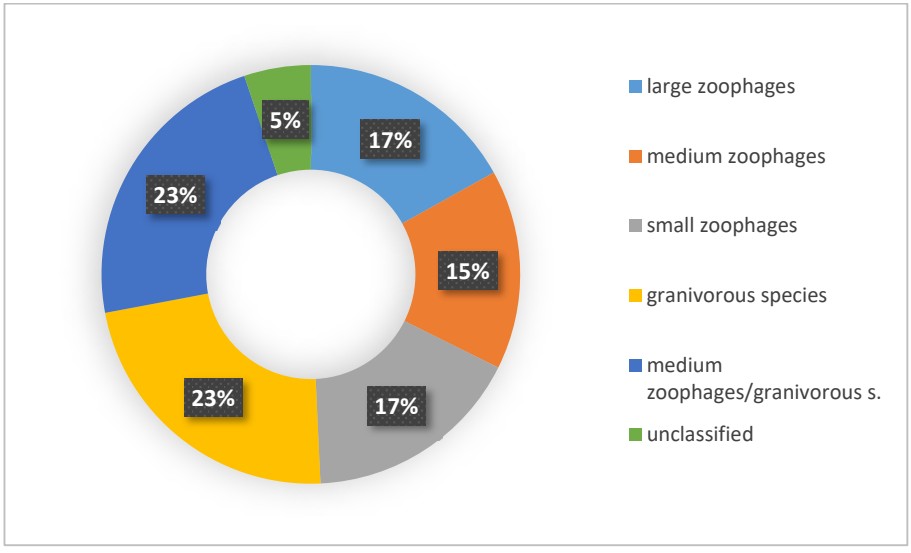

**Figure 9.** Percentage of *Carabidae* species from different trophy groups. This diagram was made based on selected publications, including data on the ecological characteristics of *Carabidae* communities inhabiting the investigated plants.

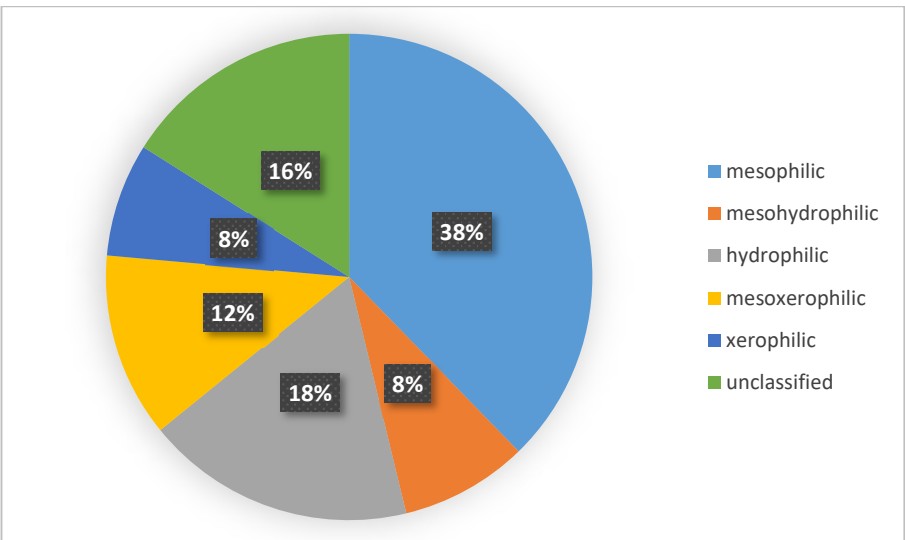

**Figure 10.** Per cent of *Carabidae* species with a different hygropreference. Diagram was made based on selected publications, including data about the ecological characteristic of *Carabidae* communities inhabiting the investigated plants.

## 4. Discussion

While the impact of willow and energy poplar plantations on abiotic factors is well-studied, there is a lack of comprehensive, multi-faceted research on the impact of plantations on biotic factors. Another problem is the lack of a homogeneous testing methodology. This makes research results difficult to interpret and compare with results obtained by other scientists. Of the many studies conducted, there were varying results, depending on the study region, the establishment phase of the SRC, its characteristics and its surroundings. Factors shaping SRC nature are adjacent land, terrain and water relations in the area where the plantation is located, the size and spatial configuration of the plantation itself, plantation age, the length of the production cycle and its phase (canopy age), and the willow strains planted. Due to a large number of variables, these results can be difficult to interpret. So far, no comprehensive research has been carried out to describe the roles of all specific factors. The impact of particular plantation features on the *Carabidae* populations inhabiting them is described below.

### 4.1. Factors Affecting Biodiversity

#### 4.1.1. Rotation Length and Canopy Age vs. Plantation Age

It was found that, in general, agricultural fields and forests were both characterized by greater species richness (estimated by the species number) than willow and poplar SRC [89]. However, it was noticed that the strictly quantitative indicator "species richness" (species numbers) is only weakly correlated with qualitative biodiversity targets such as rare and endangered or specialized species [89,96].

A factor strongly influencing the biodiversity values of SRC regarding stenotopic species is rotation length. Wegner et al. [88] hypothesized that longer rotations may favor "ubiquitous" versus "stenotopic" species in terms of anthropogenic influence being less flexible. Their research indicated that poplar plantations are the habitat preferred mainly by non-specialized species, whereas increased inflow of adaptable species is observed after the harvests and is connected with differences in the re-growth dynamics (steam sprouting is slower in older plantations, which have the characteristics of an open vegetation site during this period) [88,89]. This seems to corroborate the data from articles collected by Müller-Kroehling et al. [89], indicating that in the majority of cases, only very young SRC in the establishment phase can provide an ephemeral pioneer habitat with a

particular value for species protection [106]. Harvesting can at least partially renew a habitat function for open-habitat species by creating favorable conditions for the development of the herbaceous plant. Creating fallow-like conditions, they can act as a refugium for short-lived, open habitat species, especially granivorous or polyphagous species for which older SRC have a limited conservation value [89,92]. The analyzed data showed that not fully established plantations are a habitat for a high abundance of seed-eating species, the number of which diminishes with the age of the plantation. Similarly, in freshly cut stands, the number of granivorous species was higher, which means that harvesting promotes population increase for some of them, e.g.: *Agonum sexpunctatum, Amara plebeja, A. similata, A. aenea, Anisodactylus binotatus, Harpalus affinis, Pseudoophonus. rufipes, Poecilus cupreus* and *P. versicolor,* which are eurytopic, less specialized species. Furthermore, it was proven that there is a group of species which prefers openings and seams of SRC than strictly open habitats such as fields and grassland, particularly in highly impoverished landscapes with few ecotone habitats, as was shown for *Carabus auratus* [89]. Therefore, many authors found later rotations of SRC as having a lower biodiversity value, hence they were inhabited mostly by species assemblages dominated by common, eurytopic species [89,96].

Similarly, Müller-Kroehling et al. [89] found that the abundance of the red-listed species is negatively correlated with the age of the plantation, while for the age of stand, no significant trend was observed. Meanwhile, in-depth analysis showed that while federally red-listed species occurred mostly in the newly established SRC, for the regional lists, the trend is reversed, and the red-listed species were more abundant among older plantations. Additionally, in this case, the longitude of rotation was negatively correlated with the occurrence of the endangered species. This proves that even though willow and poplar plants are uniform regarding their age, they still undergo a directed development and the age of the plantation plays a substantially larger role than the age of the canopy attained after harvesting [87,89]. It could also be anticipated that the age of the stand was a significant occurrence factor; surprisingly, the age of the plantation was even strongly positively correlated with this species' presence [89,96]. Changes taking place during plantation growth affected microclimatic conditions, depending on other factors from the accompanying plant cover [89,107–109]. Long-established SRC plantations probably provide better soil conditions, including soil moisture and soil particle size distribution, which is important especially for the development of larval carbide forms [89,92,97,110,111]. Furthermore, as a refugium for some species endangered at the local scale, SRC may help to increase regional population stability and genetic diversity across the country. Additionally, it was found that the presence of endangered species, for example *Abax ovalis, A. carinatus* and *A. parallelus, Molops elatus* and *Carabus auratus*, was also strongly positively correlated with the proximity of forest. During this research, it was found that a plant with extended time without coppicing can serve as a refugium for forest species, contributing to their protection in the agricultural landscape. Among the collected beetles, in both qualitative and quantitative terms, dendrophile species predominated. Most of them were typically forest species, while ecotone species partly associated with tree stands were also numerous. Articles juxtaposition indicated that more than half of all identified species were characteristic for open areas, which proves the role of the energy willow plantation as a reservoir for these taxa. Such a result was also obtained by Konieczna et al. [87], who examined abandoned plantations. In this work, a large number of beetles characteristic of open areas was also observed. It is worth mentioning that the time for which the investigated plants were not harvested was similar to the period without canopy cutting on EcoSalix plantations [7,112]. Unfortunately, in some publications, significant data describing the surroundings of the plantation and its spatial arrangement as well as determining the method of sampling were lacking. This reduces the possibilities of interpretation of the data obtained. The greatest disadvantage of some works was the missing information about plantation age, which makes it impossible to determine whether the presence of forest species is the result of the canopy or the age of the plantation.

4.1.2. Surrounding Area—Environmental Corridors and Islands

The plantation area is the factor for which the influence has not been fully estimated so far. Larger plantations provide habitats for a greater amount of forest species; however, most of the research was conducted in not very large SRC [89]. Therefore, some authors claimed that large plantations can contribute to a decrease in biodiversity [96,110,113,114]. Another factor examined was the surrounding crops. Many of the endangered species are forest species with low dispersal power. Among this group, there are many stenotopic brachypterous species with reduced vestigial wings. Because of the lacking development of the hind wings, they do not have the ability for dispersal flights. Therefore, the distance to the nearest forest is, in some cases, a limiting factor [89,96,110]. Additionally, it is worth mentioning that flight ability is a species adaptation often connected with inhabiting a disturbed environment, for example, cultural land [97,110].

In high-land-pressure agricultural landscapes with high fragmentation and a lack of connectivity between individual habitats, strictly forest species can be isolated in forest "islands". According to the island theory, only large habitat patches providing enough resources can serve as a stable environment. On the other hand, vast areas of monocultures and low landscape diversity can lead to species declining, especially stenotopic species or those with high environmental requirements [115,116]. One of the main reasons for this is the genetic erosion of the population [89,96]. This issue has great importance in the SLOSS (single large or several small) debate [117–119].

It is known that green corridors prevent the isolation of the population and thus the depletion of their genetic pool [107,108]. A decrease in the effective population size leads to the disclosure of the negative effects of genetic drift and coalescence, which is more visible in small, isolated populations [109,111,113,114]. It leads to an increase in homozygosity and the loss of genetic variability. In extreme cases homogeneity can depress population fitness, general resistance to environmental factors, and flexibility in coping with environmental challenges [109,115]. Therefore, one of the assumptions of sustainable agriculture is to increase the landscape mosaicism. Based on the concept of "Islands biogeography" introduced by McArthur and Wilson [120–125], landscape corridors connecting isolated habitat patches can be applied to the agrocenoses, for example by fallowing or introducing extensive agricultural crops [37,126–128]. Establishing energy willow and poplar crops on lower-class soils or on the marginal land enables one to diversify the use of arable land, maintaining an income for local farmers [7,38]. In this type of area, the additional function of SRC, especially old ones, for forest species is being the corridor habitat [89,90,129]. The scope of this role might vary largely regionally according to inhabiting species and environmental factors. It is also worth noting that the effects of climate change will exacerbate the problem of the decline of both epigeic beetles [52,130] as well as insects in general [43,44,51,131–135].

*4.2. Carabidae as Bioindicators and Ecosystem Services Providers*

Determining the characteristics of epigeic beetle assemblies on SRC plants and making ecological descriptions of them, taking into account trophic structure, habitat preferences, hydro preferences and phenology enables their use as bioindicators [136]. Jowett et al. [137] emphasized the role of the species profile of a given population as an important factor determining habitat maturity. One of the most important features of *Carabidae* allowing us to assess the ecological status of a given environment is food preferences. Although there is no doubt that carabids have a potential for pest control [95,97,138,139–144], as well as weed seeds [145–148], there is remarkably little interest in the role they play in underpinning ecosystem services in bioenergy plantations. Rowe et al. [81] compared the processes of predation and litter decomposition in willow SRC and alternative land-uses: arable and set-aside. In the described bioassay, even though willow plants had the highest abundance and diversity of ground-dwelling arthropod predators, these factors had no detectable influence on predation rates. The reason for this was that the carabids were more active in cereal crops vs. SRC. As a result, the predation rates in the investigated plots did not differ between habitats. These observations were connected to the environmental factors, characteristic for each of the habitats. As it is known, these processes are inextricably linked to crop productivity and ecosystem

stability; however, so far, our understanding of them in SRC crops is limited, and further investigations are required [81]. Therefore, it would be very valuable to carry out research work that combines the assessment of carabid communities and the ecosystem services provided by them and takes into account environmental factors. Unfortunately, there are currently no such publications.

### 4.3. Carabidae Assemblages Structure

A detailed description of *Carabidae* groupings allows for characterizing the habitat of a young willow plantation, rich mainly in eurytopic, poorly specialized species, as a temporary environment undergoing a dynamic development [85,86]. There was a large divergence gap between the dominant and the other dominance classes, which has also been proven by other researchers (Table 2, Figure 8) and is a factor indicating the instability of the population. It was shown that the most common group were open-field carabids, characteristic for adjacent areas (Table 2, Figures 4 and 5) Walerys et al. [86] claimed that this could be the result of the small size of the plantations studied. However, such a tendency was not visible for the older plantations, most of which had a higher share of forest species (Table 2, Figure 5). In most plantations mesophilic carabids dominated [85,86], Figure10, as the most tolerant in terms of humidity requirements. Spring breeding carabids characteristic for newly colonized areas, dominated [86], Figure 7, while autumn and wintering beetle larvae, characteristic for older developmental stages [87] of stands, were absent on newly established plants. On the other hand, Kosewska et al. [85] indicated that the domination of one of the developmental *Carabidae* types in the studied area did not depend on the canopy, but was correlated with the study year [85]. Additionally, Kosewska et al. [85] showed that in older plants, beetles were less numerous than in the newly situated ones. This confirms the thesis that plantations in the initial period of the production cycle, as a transitional environment, constitute an attractive habitat for this type of species. The conducted study also showed that the number of large zoophages was higher in the older plants. This proves that the population is stable. Boháč et al. [110] investigated the impact of adjacent habitats on *Carabidae* and *Staphylinidae* population structure. As expected, the highest number of species was found on plots least affected by humans. For communities occurring in these plots, stenotopic species (e.g., *Platynus assmilis*) and large *Carabidae* species (e.g., *Carabus hortensis hortensis, C. violaceus violaceus*) were typical. Additionally, psychrophilic species characterized by winter activity were the best-represented group in these plots. Their research proved that the influence of anthropogenic factors was associated with an increased prevalence of eurytopic species, as well as an increased prevalence of species with summer imago activity.

On the other hand, forest species (e.g., *Pterostichus oblongopunctatus*), as well the protected *Carabus scheidleri*, were noted only in the fields adjacent to the forest, even though these were also fields with the highest anthropopressure. In addition, in studies evaluating carabidofauna in out of use plantations with old canopies [87], the presence of endangered and rare species was noticed. In these types of plants, the most numerous carabids were mesophilic species: *Pterostichus melanarius, P. niger* and hydrophilic *Limodromus assimilis*. *P. niger* and *L. assimilis* were classified as a forest species. Similarly, [88] reported that species preferring light forest or forest edges, such aslike *Carabus convexus*, as with species with higher humidity requirements such as the hygrophilous *Panagaeus cruxmajor*, are listed only for the longer rotations. Similar results obtained by other investigators [91] can suggest that these species are a regular component of carabidofauna in Central and Eastern Europe [87]. These results indicate the need for further research on the structure of the *Carabidae* population and prove that not the number of beetles in a given area is less important compared to their species profile.

### 5. Conclusions with Remarks

Energy willow and poplar plantations may increase the biodiversity of *Carabidae* beetles. Due to the lower use of pesticides, as well as the smaller number of agrotechnical treatments carried out, SRC can be a habitat which is more stable than annual crops, such as rape or wheat. However, energy plants also undergo dynamic changes, associated not only with the production cycle and canopy age, but also with the age of the plantation itself. This is especially important for forest species, which in

older SRC may find conditions similar to those in their natural habitat. Nevertheless, the data describing forest species requirements and biodiversity in the SRC are still lacking. It is known, however, that even if poplar and willow plants cannot provide a permanent habitat for forest species, they can serve as ecological corridors, connecting isolated land patches, inhabited by populations of epigeic insects with low dispersal powers. Energy willow and poplar plantations, if they meet several conditions, can be used for agriculture, as well as serving as a refugium for animals inhabiting areas with a high risk of anthropopression

SRC plantations can provide carabids with breeding and shelter places, important especially if the adjacent areas of arable land are characterized by a strongly intensified agricultural production. Maintaining a stable population of *Carabidae* is recommended in the integrated agriculture model, the assumptions of which include reducing the negative impact on the natural environment and human life. Ground beetles play a crucial role in providing valuable ecosystem services. Moreover, their services concern not only the energy willow plantations themselves, but also contribute to the protection of adjacent annual crops. *Carabidae* communities are still not studied enough on SRC plants. This is the result of, on the one hand, the small number of studies that have been carried out on these crops, compared to, for example, rape cultivation, and on the other hand, the fact that many works were not translated into English and appear only in local periodicals. Therefore, assessment of ground beetle population on willow and poplar plants based on community structure is currently impossible. For energy willow, there is still a lack of complete data at the regional scale, which would enable us to create the ground beetle community model describing the level of naturalness of the habitat. According to this, further investigations should be done to estimate community structure to obtain additional data on the state of the habitat and diversity of species inhabiting poplar and willow plants.

**Author Contributions:** Conceptualization, N.S.P., S.Z.C. and M.J.S.; methodology N.S.P., S.Z.C. and M.J.S.; validation, N.S.P., S.Z.C. and M.J.S.; formal analysis, N.S.P., S.Z.C. and M.J.S.; investigation, N.S.P.; resources, N.S.P., S.Z.C. and M.J.S..; data curation, N.S.P.; writing—original draft preparation, N.S.P.; writing—review and editing, N.S.P., S.Z.C. and M.J.S.; visualization, N.S.P.; supervision, S.Z.C. and M.J.S.; project administration, M.J.S.; funding acquisition, M.J.S.; All authors have read and agreed to the published version of the manuscript.

**Funding:** This paper was co-financed by the National (Polish) Centre for Research and Development (NCBiR), entitled "Environment, agriculture and forestry", project: BIOproducts from lignocellulosic biomass derived from MArginal land to fill the Gap in Current national bioeconomy, No. BIOSTRATEG3/344253/2/NCBR/2017.

**Acknowledgments:** Natalia Piotrowska is a recipient of a scholarship from the Programme Interdisciplinary Doctoral Studies in Bioeconomy (POWR.03.02.00-00-I034/16-00), which is funded by the European Social Funds.

**Conflicts of Interest:** The authors declare no conflict of interest.

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
