# Peer review of "Ground Beetles (Carabidae) in the Short-Rotation Coppice Willow and Poplar Plants—Synergistic Benefits System"

_agriculture, doi:10.3390/agriculture10120648_

Round 1
Reviewer 1 Report
The authors have raised very important question in their investigation – short rotation coppice (SRC) plantations as environmental islands that can be refugia and environmental corridors for endangered populations. The acreage of lignocellulosic plants is increasing now and they are important in heat and power generation and in the second-generation liquid biofuels production. The aim of the study was to estimate biodiversity and abundance parameters in SRC plantations taking into account ground beetles.
Despite the comprehensiveness of the study, I have the following major notes, as I review the Review article:
- Nearly 200 articles are presented in References, but in the only 9 ones (85-93) we find the data concerning the discussed item – ground beetles in SRC plantations.
- Geography of paper studied seems very narrow. There are a lot of investigations on the topic of discussion in other countries (e.g. Russian Federation) and there were not taken into account.
- Incorrect classification of the ground beetles. Line 85 – 88. Classification should contain separate paragraphs: size (large, medium, small), nutrition (zoophagues, hemizoophagues, herbivores), life-cycle features (spring time-, autumn-time breeders), habitat preferences, humidity preferences etc.
Else. Line 282. “Ubiquitous” and “adaptable” are synonyms unlike “stenotopic” species.
- Table 1. Column “Adjacent habitats. Practically in a half of the investigations this parameter was not studied. At the same time it is one of the main ones when describing carabids community structure in the given biotope (see Matalin AV, Makarov KV (2011) Using demographic data to better interpret pitfall trap catches. In: Kotze DJ, Assmann T, Noordijk J, Turin H, Vermeulen R (Eds) Carabid Beetles as Bioindicators: Biogeographical, Ecological and Environmental Studies. ZooKeys 100: 223–254. doi: 10.3897/zookeys.100.1530): often the sampled beetles are the immigrants from the surrounding lands but not the permanent element of the communities.
- The authors declare themselves that in works they have discussed there is a lack of comprehensive multi-faceted research and their results are incomparable (line 209 – 210).
- Results of the works, mentioned in Results, sometimes contradictory (see 87, 89), so it is impossible to draw unambiguous conclusions about SRC impact on ground beetles communities.
- Section 3.2.3. “Population structure” implies the structure of definite species populations. In this section communities (assemblages) structure is discussed.
Author Response
- Nearly 200 articles are presented in References, but in the only 9 ones (85-93) we find the data concerning the discussed item – ground beetles in SRC plantations.
Response 1: In the review publication, we have used the articles listed in the largest Scopus and Web of Science databases and in one regional database, appropriate for the country of origin of the authors. These databases did not show any more publications on the topic. To increase the number of articles, we also used the literature from the publications we cited. List of publications extracted from the bibliography, related to the topic, and not used by us when creating the article in the form of an Excel file. Unfortunately, it was impossible to use more publications for several reasons:
- Most of these publications are placed only in journals and with low or no IF, therefore they do not end up in public databases; some publications of this type also include doctoral dissertations, not published on the web;
- Publications are unavailable due to language barrier (Italian, Czech, German etc.)
- These publications contain too little data to be used for this type of compilation - the data is incomplete or inadequate;
Additionally, in comparison with e.g. publications on wheat or rape, the amount of literature is relatively small. This proves that the topic requires further research.
- Geography of paper studied seems very narrow. There are a lot of investigations on the topic of discussion in other countries (e.g. Russian Federation) and there were not taken into account.
Response 2: As in the first point - these publications are not available in large databases. Our original idea was to search for publications mainly from Central and Eastern Europe, but due to the language barrier obtaining information from local databases was not possible. In addition, a large part of the Russian Federation and Scandinavia have been excluded by us due to the climatic conditions in their area.
- Incorrect classification of the ground beetles. Line 85 – 88. Classification should contain separate paragraphs: size (large, medium, small), nutrition (zoophagues, hemizoophagues, herbivores), life-cycle features (spring time-, autumn-time breeders), habitat preferences, humidity preferences etc.
Else. Line 282. “Ubiquitous” and “adaptable” are synonyms unlike “stenotopic” species.
Response 3: In this work, the classification was established on the basis of the division taken from the source works, so that the extraction of as many data as possible was possible. I consider the comment of the reviewer a valuable hint - including the size and trophic preferences as one parameter may reduce the possibilities of analysing Carabidae communities. We will certainly use this classification as a more accurate one in the research work that is currently underway. In the review, however, we will stick to the alternative classification used by the authors of the source works.
Line 282 has been corrected.
- Table 1. Column “Adjacent habitats. Practically in a half of the investigations this parameter was not studied. At the same time it is one of the main ones when describing carabids community structure in the given biotope (see Matalin AV, Makarov KV (2011) Using demographic data to better interpret pitfall trap catches. In: Kotze DJ, Assmann T, Noordijk J, Turin H, Vermeulen R (Eds) Carabid Beetles as Bioindicators: Biogeographical, Ecological and Environmental Studies. ZooKeys 100: 223–254. doi: 10.3897/zookeys.100.1530): often the sampled beetles are the immigrants from the surrounding lands but not the permanent element of the communities.
Response 4: Unfortunately, we do not have access to data from a few years ago determining the use of the area adjacent to the surveyed plantations. This has been described as a factor in one of the main problems in the description of Carabidae communities in this review. In the current research work, this factor will be taken into account, in line with the valuable suggestion of the reviewer. Unfortunately, we have no influence on the lack of this data in the source publications.
- The authors declare themselves that in works they have discussed there is a lack of comprehensive multi-faceted research and their results are incomparable (line 209 – 210).
Response 5: This point is related to point 4. - unfortunately, the source publications lack data such as: adjacent environment, age of the stand, age of plantations. Unfortunately, due to the small number of available publications on the topic under study, we were forced to use also those that lack information. We emphasized this deficiency in order to draw attention to the need to include complete data in future publications, so as to develop a common standard enabling the analysis of various works in a similar way.
- Results of the works, mentioned in Results, sometimes contradictory (see 87, 89), so it is impossible to draw unambiguous conclusions about SRC impact on ground beetles communities.
Response 6:
As noted by the reviewer, in some cases it is not possible to make an unequivocal conclusion about the impact of the SRC on Carabidae assemblages. In our review, we presented the currently known views on this problem, presented by various researchers. However, more research is needed to draw firm conclusions about this effect.
- Section 3.2.3. “Population structure” implies the structure of definite species populations. In this section communities (assemblages) structure is discussed.
Response 7:
Corrections were made in accordance with the reviewer's recommendation.
Thank you kindly for reviewing the text and valuable comments, which will surely help to improve my work in the future.
Reviewer 2 Report
The paper is a review of some significant articles about the diversity of carabid beetles in willow and poplar energy plantations selected amond a large variety of european papers found through some scientific databases. Apart some problems with english (I enclose a copy of the PDF with some corrections, but the entire manuscript should be raed by a native speaker), I found the structure of the artcile too heavy and with too speculative parts inside the results. Paragraphs like 3.2 Factors concerning biodiversity don't concern results gained by the authors but simply refer on facts and guidelines already published by other auhors, also some subsequent parts enclosed in the results are too rich of comments about the data, they should be strongly shortened and restricted to the data elaborated by the authors! Several concepts are repeatedly found both in the introduction, where they are ok, as well as in the results.
On the whole, the results should contain exclusively the elaborations done by the AA. Concerning the contents of the data presented, it seems that too many carabid species hve been considered as "unclassified", despite some european databases are rich in information, as Homburg et al. Insect Conservation and Diversity, 2013, doi: 10.1111/icad.12045, and should be used for less known species.
The paper is anyway an useful synthesis, it seems written in at least two phases, because in the introduction some old terms like hemizoofages are used, whereas in the results the term granivorous, now more used, is found.
I tried to insert into introduction some general citations about carabid ecology that cannot be omitted, and I encourage the authors to shorten the redundant parts and to resubmit the article.
Also Table 1 is very hard to read and could be restructured with smaller letters, better separating uno paper from teh next.

Author Response
As recommended by the reviewer, we shortened the results by removing unnecessary fragments, including the entire Paragraph 3.2. and a large section containing descriptions of the results of other researchers. Fragments are left that will allow the reader to refer to the results obtained by other researchers without the need to refer to source works.
Wherever possible, we supplemented the ecological descriptions of carabids, using the www.carabids.org database from the article proposed by the reviewer and from the Polish database https://baza.biomap.pl. This resulted in the need to correct the charts and the references in the text, which was done.
The table has been corrected and the bibliography has been extended with suggested articles, additionally all corrections suggested in the attached PDF file have been introduced - they can be traced in the attached manuscript.
Thank you very much for taking the time to review our article. I consider all the corrections very valuable, they will definitely contribute to the improvement of my work in the future.

Round 2
Reviewer 1 Report
I am satisfied with authors replies to my Reveiw
Author Response
Thank you very much for your insightful review
Reviewer 2 Report
I'm very sorry to see that the MS has been only very little shortened, there are still entire paragraphs that should be omitted because fully aoutside the aim of the paper, as for example 3,2,4, where results about fully different kind of crops are reported, stating sinply that similar studies have till now not made for SRC. Such paper organization can be accepted only in a graduate thesis but not in a scientific journal.
I hearthly invite the authors to remain in the subject and rewrite the results separating them from the discussion. I have seen that some diagrams have been enhanced by a better analysis of the food preferences and ecology, but on the whole the matter of the metastudy is little organized and could be highly improved by adding some statistics to the analysis.

Author Response
Thank you very much for your insightful review and for pointing out some issues with Carabidae research. These are inspirational comments that we will use in future research and analysis based on our field research. However, we dare to argue with some of the Reviewer's statements.
I'm very sorry to see that the MS has been only very little shortened, there are still entire paragraphs that should be omitted because fully outside the aim of the paper, as for example 3,2,4, where results about fully different kind of crops are reported, stating simply that similar studies have till now not made for SRC.
We dare to disagree with the Reviewer. The indicated fragments fully fit into the aim of the paper, marked in the preliminary.
“The novelty of this work is it showing a new view on SRC plantations as environmental islands - areas that can be refugia and environmental corridors for endangered populations. The purpose of this work is not only to show the diversity of entomofauna, but also to draw attention to the relationship between the habitat of Carabidae and the shape of their population and their ecologic function, which investigations has not been extensively developed so far. This will allow a better understanding of the role not only of the ground beetles itself, as a group providing ecosystem services, but also of the entire complex environment of the energy willow plantation,”
Our work has a review character. It relates entirely to the SRC. In the review of papers, we have indicated an important problem and an important factor determining the biodiversity of the SRC. This factor is the environmental effect, well described in the island ecology model. For this reason, it is advisable to indicate the need to analyze the areas adjacent to the SRC. We emphasized the importance of this factor in subsection 3.2.2. (Surrounding area – environmental corridors and islands).
Chapter 3.2.4 is very important for this paper and deleting it would significantly reduce the quality of the article. Carabidae are bioindicators of the state of the environment. Based on the literature review, we clearly indicate the lack of any analyzes with regard to SRC and indicate the need for research in this direction. This is an important observation of the review work. In addition, we point out the need for analysis in the context of the neighborhood and ecology of islands, and Figure 11 contained in this subchapter is a generalization and outlining the necessary research - an indication of what factors and aspects are worth paying attention to in future research. Identifying white spots in research is an important feature of reviews.
“The results of this investigations showed that energy SRC plants and Carabidae communities can create a synergistic system of mutual benefits. Willow and poplar plants can be a valuable habitat due to the increased biodiversity of entomofauna. Additionally SRC create a transitional environment that allows insects migration between isolated populations. On the other hand, ground beetles are suppliers of ecosystem services and have a significant contribution to the building sustainable agriculture by pest control, thereby ameliorating damage to field crops.”
I hearthly invite the authors to remain in the subject and rewrite the results separating them from the discussion.
This work is a review based on the available literature. In the methodological part (Materials and Methods / Data collection and Selection), we indicated how we got to these works and pointed out the shortcomings in database searches (not all works can be found in this way, and many of them had to be rejected due to unsuitability e.t.c.). Detailed presentation of the methodology is important for the readers, who would like to verify the results of our research and reach other publications not included in our work.
I have seen that some diagrams have been enhanced by a better analysis of the food preferences and ecology, but on the whole the matter of the metastudy is little organized and could be highly improved by adding some statistics to the analysis.
We will use the right and valuable comments of the Reviewer regarding in-depth statistical analyzes in another work, based on the results of our field research (they will be completed in September 2021). Then it would be most reasonable to use a full Carabidae analysis based on the different ecological classifications of all species. Wherever possible, we will try to use for statistical analysis data contained in other publications, not only related to the SRC. We will use the Reviewer's valuable advice in developing our own field research and preparing publications with the results of our own field research